# ICE: Image-Caption Encoding for Improved Out-Of-Distribution Generalization In Vision-Language Models

## Abstract

Recent advances in vision-language models have combined contrastive approaches with generative methods to achieve state-of-the-art (SOTA) on downstream inference tasks like zero-shot image classification. However, one persistent issue of these models for image classification is their out-of-distribution (OOD) generalization capabilities. We first show that when an OOD datapoint is misclassified, the correct class can be typically found in the Top-$K$ predicted classes. In order to steer the model prediction toward the correct class within the top predicted classes, we propose the *Image-Caption Encoding (ICE)* method, a straightforward approach that directly enforces consistency between the image-conditioned and caption-conditioned predictions at evaluation time only. Intuitively, we take advantage of unique properties of the generated captions to guide our local search for the correct class label within the Top-$K$ predicted classes. We show that our method can be easily combined with other SOTA methods to enhance Top-1 OOD accuracies by *0.5% on average* and *up to 3% on challenging datasets*.

## 1 Introduction

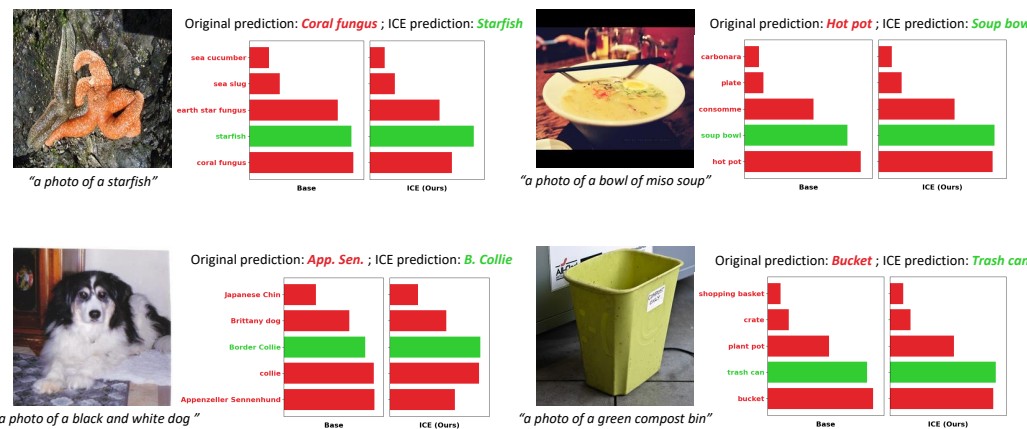

Figure 1: A demonstration for how our ICE method can be used to reclassify correctly. In these examples, ICE is applied directly to a frozen pre-trained CoCa model for zero-shot classification. Using the contexts given from the generated captions, ICE is able to successfully influence the pretrained model into predicting the correct classes.

There has been rapid progress in zero-shot image classification over the past two years, thanks to advancements in vision-language (VL) pre-training such as CLIP, ALIGN, and BLIP (Radford et al., 2021; Cohen, 1997; Li et al., 2022). At a high level, these models contain a pair of encoders that project visual and textual inputs into a joint latent embedding space. As described in the CLIP framework (Radford et al., 2021), zero-shot classification can be reformulated as an image-to-text retrieval problem, where the class name closest to the image in embedding space is predicted as the label. However, state-of-the-art (SOTA) zero-shot classification lags behind in-distribution supervised fine-tuning on all benchmarks. In many applications, in-distribution data is not available during training, so fine-tuning on out-of-distribution (OOD) source data in a way that generalizes to unseen data and labels remains an important problem. Prior works in this area follow two general

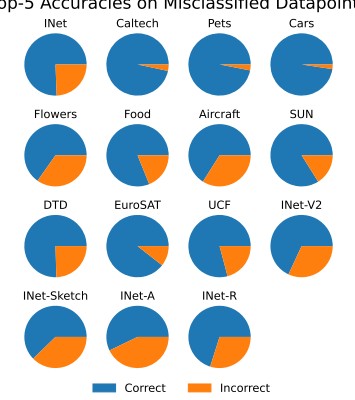

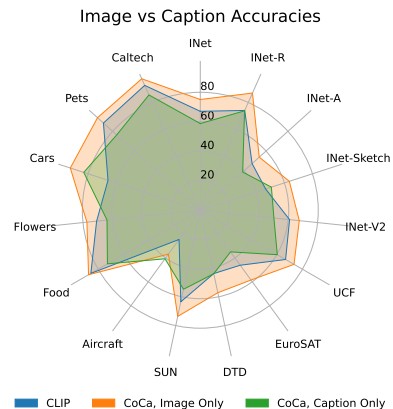

(a) A visualization of the Top-5 accuracies on misclassified Top-1 datapoints in each test dataset. Recall that correct Top-5 classifications form a strict superset over the correct Top-1 classifications. We observe that across all datasets, the true correct class can be found within the Top-5 predicted classes for most misclassified datapoints.

(b) A visualization of Top-1 accuracies between CLIP, CoCa using image embeddings only, and CoCa using caption embeddings only. We observe that while caption embeddings generally underperform compared to standard CoCa, they still retain competitive performance. We include more details on datasets and experiments in Section 4.

Figure 2

directions: (1) *Few-shot OOD* methods such as CoOp (Zhou et al., 2022b), CoCoOp (Zhou et al., 2022a), and MaPLe (Khattak et al., 2023) fine-tune the VL model on generic few-shot source data (e.g. ImageNet). The fine-tuning process is constrained to a carefully selected subset of parameters to ensure generalization to target datasets. (2) *Zero-shot* methods, such as Menon & Vondrick (2022) and manual prompt ensembling (Radford et al., 2021), focus on refining the zero-shot prediction without additional fine-tuning. These methods do not require additional data, but they typically either require a large closed-source LLM or human-engineered prompts.

*Our goal is to contribute to the zero-shot classification literature by leveraging captioners, which is previously under-explored.* Towards this goal, we first observe in Figure 2a that the Top-$K$ accuracy (the percentage of samples where the correct label is within the $K$ classes with highest predicted scores, $K > 1$) is consistently higher than the Top-1 accuracy. The reason is that the Top-$K$ predicted classes form a strict superset of the Top-1 predicted classes when $K > 1$, and thus characterize a wider range of potentially correct classes. We observe from Figure 2a that when an image is misclassified, the correct class can usually be found within the Top-5 predicted classes. Thus, our motivating question is: in order to improve Top-1 accuracy, *how can we steer the model prediction toward the correct prediction within the Top-$K$ predicted classes?*

Before we address this question, we first note that current SOTA zero-shot image classification methods perform a nearest-neighbor search between image and text CLIP embeddings (Radford et al., 2021). Recently, Yu et al. (2022) proposed CoCa, which extends CLIP with an additional text decoder. This text decoder is trained to output a description of the image by cross-attending to *all* image tokens outputted by the image encoder. Consequently, the decoder output captures fine-grained spatial information that may be absent from the image cls token. Furthermore, the caption verbalizes the content of the image as discrete text tokens, which can oftentimes be used to directly infer the image label. These advantages are illustrated in Figure 2b, where we use a spider plot to compare the Top-1 zero-shot accuracies achieved by CLIP image embeddings, CoCa image embeddings, and CoCa caption embeddings, across 15 datasets. We notice that while the caption-only CoCa under-performs compared to standard CoCa, it is still competitive and even surpasses CLIP on many datasets. This means that the captions must contain enough information about the image to supplement the standard CoCa zero-shot prediction.

To leverage this additional information, we propose a novel zero-shot method called *Image-Caption Encoding (ICE)*, where we combine the information encoded by both image embeddings and caption embeddings in order to make a more informed decision at test time. As seen in Figure 3, ICE is a

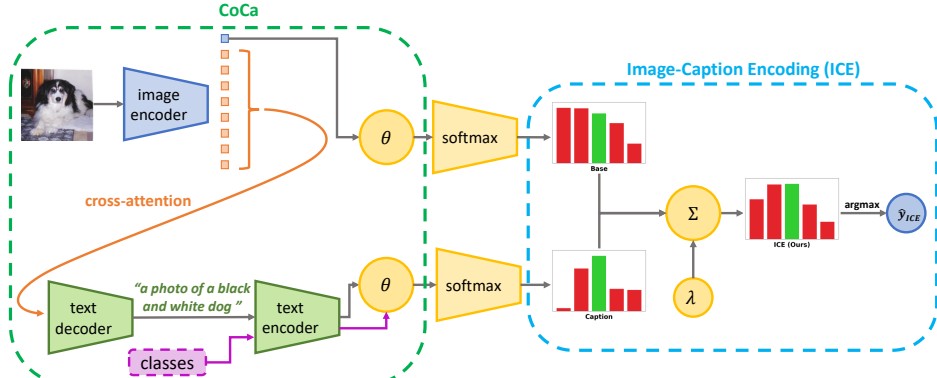

Figure 3: An overview of our *Image-Caption Encoding (ICE)* method. The diagram outlined in green describes the standard CoCa (Yu et al., 2022) workflow for obtaining the image and class embeddings. Here, we also query CoCa for the caption, and obtain the caption embedding using the text encoder. We calculate the image and caption probability distributions over the classes by passing the image embeddings, caption embeddings, and class embeddings through the cosine similarity function $\theta$ and softmax operation. Then, we select the Top-$K$ classes and perform a weighted sum of the image and caption probabilities. The weight on the caption prediction $\lambda$ is adaptively selected based on the relative confidence of the image and caption predictions.

zero-shot method with no training component and can be easily paired with existing SOTA methods for improved downstream classification performance.

Although ICE draws inspiration from traditional ensembling techniques, there are several key differences. First, ICE leverages the predictions obtained from a *single* model rather than those from several *different* models. Second, instead of aggregating predictions over all classes, we only consider the Top-$K$ image predicted classes. Third, we incorporate an innovative confidence selection mechanism that sets the weight on the caption prediction dynamically. Finally and most importantly, we exploit specific properties induced within the captions by a pre-trained CoCa model that are not present in the image embeddings for standard zero-shot classification. We discuss these properties in-depth in Section 3.4 and demonstrate specific examples of their non-trivial impact in Section 4.3.

Our contributions are as follows:

1. We propose *Image-Caption Encoding (ICE)*, a novel zero-shot classification method that utilizes information from both images and captions to make a decision at evaluation-time only.

2. We provide experimental results for ICE paired with several different SOTA baselines across 15 different OOD datasets. We show consistent improvements of $0.5\%$ on average and up to $3\%$ on several challenging datasets.

3. We analyze the benefits and drawbacks of using ICE, and provide ablation studies to analyze the effects of changing different parameters in our ICE method.

## 2 RELATED WORKS

**Multimodal foundational models**. Many VL foundational models have emerged over the past two years, including CLIP (Radford et al., 2021), ALIGN (Cohen, 1997), BLIP (Li et al., 2022), and CoCa (Yu et al., 2022). These models achieve SOTA on VL tasks by using vast quantities of un-curated image-text data from the web. CLIP uses an image encoder and a text encoder to project the two modalities into a joint latent embedding space. Popular downstream applications include zero or few-shot classification and image-text retrieval. CoCa improves CLIP by additionally training a text decoder to explain the embedding space with a caption. CoCa achieves SOTA in zero-shot classification and retrieval, but can also perform VL understanding tasks such as captioning and VQA with minimal additional supervision. In the current work, we leverage the captioning capability of CoCa for robust classification.

**Robust fine-tuning**. There is growing interest in fine-tuning multimodal foundational models on limited training data such that the resulting model achieves high accuracy even on domain-shifted

data and data with labels not seen during training. Many modern approaches rely on prompt tuning and ensembling. CoOp (Zhou et al., 2022b) is a seminal work which treats the prompt in front of the label names as soft learnable tokens. CoCoOp (Zhou et al., 2022a) trains a meta-network to condition the prompt tokens on the image embedding. MaPLe (Khattak et al., 2023) shows that learning a conditional visual prompt jointly with textual prompts improves target accuracy. All three works achieve impressive results on a diverse set of test datasets despite only being trained on few-shot ImageNet data. ClipOOD (Shu et al., 2023) uses an adaptive margin loss to optimize the visual encoder only, attaining good results on domain generalization benchmarks. Our proposed method ICE is a training-free approach that can be readily combined with the above fine-tuning methods to yield higher accuracy on most target datasets.

**Ensembling for robust classification**. Ensembling methods leverage multiple diverse predictions to form a robust final prediction; many recent works, including ours, focus on discovering new sources of diversification. WiSE-FT (Wortsman et al., 2022) calculates a weight space ensemble of the fine-tuned and pre-trained models to increase robustness under distribution shifts in target data, while inference time remains the same. Vogt-Lowell et al. (2023) demonstrates combining cross-entropy fine-tuning with stochastic weight averaging improves domain generalization. Menon & Vondrick (2022) use GPT descriptions to generate a more diverse set of text prototypes for zero-shot classification. Radford et al. (2021) use an ensemble of 80 handcrafted prompts to achieve the same goal. In our paper, while drawing inspiration from ensembling, we introduce the novel use of CoCa captions as a unique source of diversification, an avenue not yet explored by previous studies. Importantly, our approach is designed to seamlessly integrate with other zero-shot methods, as demonstrated in our experimental results.

## 3 METHODOLOGY

### 3.1 PRELIMINARIES

Consider a dataset $\mathcal{D} \subset \mathcal{I} \times \mathcal{T}$ where $\mathcal{I}$ is the image domain and $\mathcal{T}$ is the text domain, and $(I_i, T_i)$ forms a corresponding image-text pair (i.e. $T_i$ is a caption that describes $I_i$). In the CLIP framework (Radford et al., 2021), there are two main architectural components: an image encoder $f_{\mathcal{I}} : \mathcal{I} \to \mathbb{R}^l$ that maps images to a shared latent space, and a text encoder $f_{\mathcal{T}} : \mathcal{T} \to \mathbb{R}^l$ that maps text to the same shared latent space. Both encoders are pre-trained using a contrastive loss that pulls corresponding image-text embeddings close together in latent space, and pushes non-corresponding image-text embeddings away from each other in latent space. CoCa (Yu et al., 2022) extends CLIP with an additional text decoder, trained using a next-token-prediction loss, to provide captioning functionality. During image classification, only the CLIP component of the CoCa architecture is considered.

In image classification following the CLIP framework (Radford et al., 2021), for an image vector $I$ and class labels vector $y := [y_1, y_2, \ldots, y_m]^\intercal$, we first feed each class label through a prompt skeleton to obtain class prompts (e.g. for class "cat" and a prompt skeleton "A photo of a {}", the resulting prompt is "A photo of a cat"). Then, both the image and class prompt vectors pass through the image and text encoders to obtain latent embeddings $\widetilde{I}$ and $\widetilde{y}$, respectively. The predicted class label $\hat{y}$ is then given by $\arg\max_i \theta(\widetilde{I}, \widetilde{y}_i), i \in \{1, 2, \ldots, m\}$, where $\theta : \mathbb{R}^l \times \mathbb{R}^l \to \mathbb{R}$ is the cosine similarity function.

### 3.2 IMAGE-CAPTION ENCODING

Consider a text decoder $f_{\phi} : \mathcal{P} \times \mathcal{I} \to \mathcal{T}$ that maps a prompt $p \in \mathcal{P}$ (e.g. "a photo of") and an image $I$ to a caption $c \in \mathcal{T}$. We can feed caption $c$ back through the text encoder to obtain $\widetilde{c} := f_{\mathcal{T}}(c)$.

Using the Softmax function (Bridle, 1989), we obtain the class probabilities for image $I$ and caption $c$, respectively, as

$$S^I := \text{Softmax}\left(\left[\theta(\widetilde{I}, \widetilde{y}_1), \theta(\widetilde{I}, \widetilde{y}_2), \ldots, \theta(\widetilde{I}, \widetilde{y}_m)\right]^\intercal\right)$$

$$S^c := \text{Softmax}\left(\left[\theta(\widetilde{c}, \widetilde{y}_1), \theta(\widetilde{c}, \widetilde{y}_2), \ldots, \theta(\widetilde{c}, \widetilde{y}_m)\right]^\intercal\right). \tag{1}$$

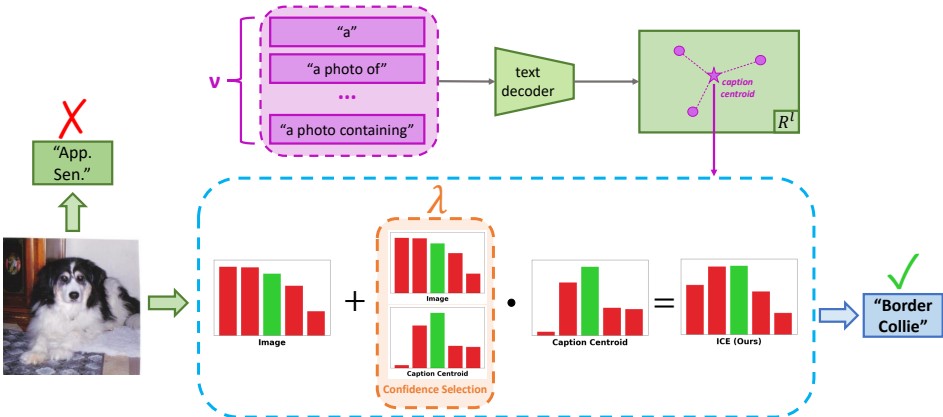

Figure 4: A more detailed look at how our *Image-Caption Encoding (ICE)* method works. In practice, instead of using a single caption for ICE, we use the centroid of $\upsilon$ differently-prompted caption embeddings. Then, using the centroid caption embedding, we adaptively select the $\lambda$ weight by comparing the standard deviations of the image prediction probabilities and caption prediction probabilities, over the Top-5 classes. The final ICE scores are then a $\lambda$-weighted sum between the two probability distributions.

The indices corresponding to the $K$ classes with highest image-predicted probability, and the final ICE prediction, are computed respectively as:

$$\Omega_K^I := \underset{J \subset M, |J|=K}{\arg \max} \sum_{j \in J} S_j^I \qquad\qquad \underset{\omega \in \Omega_K^I}{\arg \max} \, S_\omega^I + \lambda S_\omega^c \qquad (2)$$

where $M = \{1, 2, \ldots, m\}$ and $\lambda$ is a scalar variable. In essence, the ICE prediction remains anchored within the primary Top-$K$ classifications as determined by the image class probabilities. When we incorporate the caption scores tied to these Top-$K$ predictions, it reshapes the probability landscape over the initial Top-$K$ image-determined classes. By selecting the class with the highest probability from this refined distribution, we aim to align closer with the true class while retaining the accuracy of previous classifications. Intuitively, the caption probability distribution should provide information about the image that is not clear or fully captured by the image probability distribution, as detailed in Section 3.4, and their aggregated prediction should provide a more reasonable downstream prediction. Figures 3 and 4 contain a high-level overview and an in-depth visual interpretation of our method, respectively.

### 3.3 Additional Modifications

As visualized in Figure 4, in our experiments, we use the centroid of a diverse set of captions rather than a single caption for increased robustness. That is, for a set of prompts $P \in \mathcal{P}^\upsilon$, we generate a set of corresponding captions $C \in \mathcal{T}^\upsilon$, obtain their caption embeddings $\widetilde{C} := \{f_\mathcal{T}(c_1), f_\mathcal{T}(c_2), \ldots f_\mathcal{T}(c_\upsilon)\}$, and finally, their centroid $\widetilde{\overline{c}} := \frac{1}{\upsilon} \sum_i^\upsilon \widetilde{c}_i$. The centroid $\widetilde{\overline{c}}$ is then used in place of $c$ in Equations 1 and 2.

In addition, we dynamically compute the caption scalar variable $\lambda \in \mathbb{R}_+$ as a function of the standard deviation of the captions. That is, given some image $I$ and caption $c$, we compute $\lambda$ as

$$\lambda = \xi \frac{\sigma(S_K^c)}{\max(||[\sigma(S_K^I), \sigma(S_K^c)]||_2, \epsilon)} \qquad (3)$$

where $\xi$ is a constant, $\epsilon$ is a small constant, $S_K^I$ and $S_K^c$ are the Top-$K$ probabilities for $S^I$ and $S^c$ respectively (i.e. the probabilities whose indices are specified by $\Omega_K^I$), and $\sigma$ is the standard deviation operator. Intuitively, the standard deviation of the Top-$K$ highest probabilities of the image and caption distributions correspond to the model confidence about their respective predictions. On one hand, when the model confidence for the image prediction is high and the caption confidence is low, then the caption probabilities should not influence the image probabilities as much. On the other hand, when the image confidence is low but the caption confidence is high, the caption probabilities

should more heavily influence the image probabilities. In the event when both image and caption confidences are high or low, the default weighting would be relatively equal. The constant $\xi$ can specify how much the caption probabilities should affect the image probabilities overall.

### 3.4 CAPTION PROPERTIES

In general, the zero-shot accuracy using only the caption embeddings is significantly (on average about 5 %, see Table 1) lower than the zero-shot accuracy using image embeddings, with the notable exception of aircraft fine-grained classification. Caption-only zero-shot classification is often unreliable, since the caption does not always correspond to one of the class names. For example, a picture containing a teddy bear on top of a bed might be captioned as "a picture of a teddy bear", ignoring the bed in the background. However, if teddy-bear is not one of the labels, and the correct label is "bed", the caption does not provide useful information for the classification problem. For this reason, the optimal hyperparameters for ICE place a greater emphasis on the prediction from the image embedding $\widetilde{I}$. Nonetheless, for the caption embedding to contribute to a higher aggregate accuracy, we only require that the caption-predicted probabilities $S_K^c$ be not completely correlated with $S_K^I$. In other words, *the caption sometimes contains extra information that nudges the prediction in the correct direction*. We list here a few intuitions for why using captions can improve overall classification accuracy:

1. The CoCa text decoder cross-attends to all output image tokens from the vision encoder, while the image prediction only uses the output cls token. The image token matrix contains spatial information that may be pertinent to the classification problem.

2. The CoCa text decoder was trained to output web image captions with a language-modeling loss. Consequently, CoCa exhibits some rudimentary reasoning ability based on memorization of relationships between certain concepts. For example, the text decoder has learned that the painting "the starry night" is authored by Vincent van Gogh. This correspondence is memorized by the weights of the decoder and may be useful for some classification problems. In our experiments, we found that the caption prediction is much better than the image prediction on aircraft classification. This is likely because the correspondence between fine-grained visual concepts and the aircraft model name is encoded in the text decoder.

3. The CoCa caption effectively isolates visual concepts; this is an inherent property of textual data. For example, a caption that reads "a rough red blanket" effectively isolates the texture, color and content of the image. In our experiments, we found that captions on the EuroSAT dataset often isolate the land-use information from the geographical information, e.g. "a photo of agricultural land in China". The caption explicitly separates useful information (agricultural land) from information that is irrelevant to the classification problem (China). Consequently, a classifier trained on captions is less likely to learn domain-specific spurious correlations, especially in the few-shot setting.

We provide concrete examples of these discussed intuitions in our empirical analysis in Section 4.3.

## 4 EXPERIMENTS

In our experiments, we analyze the impact of combining ICE with different baselines across a suite of benchmarks. We show that our method can give consistent improvements without requiring additional training. In addition, we analyze several data points to show how ICE improves over the base method that it is paired with. All implementation details can be found in Appendix A.

**Datasets.** Our datasets are split between two common OOD categories: *cross-dataset generalization* and *domain generalization*. For cross-dataset generalization, each evaluation dataset has mostly non-overlapping classes and unrelated data distributions for zero-shot classification. For domain generalization, the evaluation datasets are domain-shifted variations of the ImageNet dataset and share the same classes as ImageNet. We evaluate our method on 11 cross-dataset generalization datasets covering a wide range of image recognition tasks. These include two generic objects datasets, ImageNet (Russakovsky et al., 2014) and Caltech101 (Fei-Fei et al., 2004); five fine-grained datasets, OxfordPets (Parkhi et al., 2012), StanfordCars (Krause et al., 2013), Flowers102 (Nilsback & Zisserman, 2008)), Food101 (Bossard et al., 2014), and FGVCAircraft (Maji et al., 2013); a scene categorization dataset, SUN397 (Xiao et al., 2010); an action recognition dataset, UCF101 (Soomro et al., 2012); a describable textures dataset, DTD (Cimpoi et al., 2013), and a satellite images dataset, EuroSAT (Helber et al., 2017). In addition, we consider four domain gener-

| | | Cross-dataset Evaluation Targets | | | | | | | | | | | Domain Generalization Targets | | | | |
|---|---|---|---|---|---|---|---|---|---|---|---|---|---|---|---|---|---|
| | INet | Caltech | Pets | Cars | Flowers | Food | Aircraft | SUN | DTD | EuroSAT | UCF | Average | INet-V2 | INet-Sketch | INet-A | INet-R | Average |
| Zero-shot(Image) | 75.1 | **97.6** | **93.8** | 92.7 | 77.3 | 87.5 | 36.8 | 73.6 | 57.2 | 58.5 | 73.4 | 74.8 | 67.5 | 63.5 | 53.8 | 87.0 | 68.0 |
| Zero-shot (Caption) | 58.8 | 85.6 | 76.3 | 83.1 | 63.6 | 72.7 | **40.7** | 54.8 | 44.0 | 34.9 | 60.3 | 61.6 | 50.2 | 50.7 | 38.7 | 73.8 | 53.3 |
| + ICE | **75.6** | 97.1 | **93.8** | **93.0** | **78.0** | **87.7** | 38.3 | **74.0** | **59.3** | **61.3** | **74.3** | **75.7** | **67.8** | **64.0** | **54.4** | **87.5** | **68.4** |
| Manual Prompts | 75.5 | **97.1** | 93.7 | 92.7 | 77.5 | 87.5 | 37.5 | 74.0 | 60.7 | 60.2 | 73.8 | 75.5 | 67.8 | **64.7** | 53.2 | 88.1 | 68.4 |
| + ICE | **75.9** | **97.1** | **93.8** | **93.1** | **78.7** | **87.6** | **39.7** | **74.1** | **61.6** | **61.2** | **73.9** | **76.1** | **68.2** | **64.7** | **54.4** | **88.4** | **68.9** |
| GPT Centroids | 74.8 | **97.8** | **93.3** | 92.4 | 75.8 | 87.4 | 36.4 | 73.9 | 58.8 | 63.7 | 73.2 | 75.3 | 67.3 | 63.3 | 52.6 | 86.7 | 67.5 |
| + ICE | **75.2** | 97.4 | 93.0 | **92.8** | **76.3** | **87.7** | **39.1** | **74.2** | **60.2** | **64.2** | **73.7** | **75.9** | **67.4** | **63.5** | **53.5** | **87.1** | **67.9** |
| GPT Score Mean | 74.9 | **97.6** | **93.7** | 92.4 | 76.2 | 87.3 | 36.2 | 73.9 | 58.9 | 64.9 | 73.6 | 75.5 | 67.6 | 63.5 | 52.8 | 86.8 | 67.7 |
| + ICE | **75.4** | 97.4 | 93.5 | **92.8** | **77.0** | **87.6** | **39.2** | **74.2** | **60.2** | **65.5** | **73.9** | **76.1** | **67.9** | **63.7** | **53.4** | **87.2** | **68.1** |

Table 1: Comparison with *zero-shot baselines* on 15 test datasets. We always observe that stacking our ICE method on top of baseline methods provides consistent improvements. All methods are zero-shot and use CoCa ViT-L/14. The caption zero-shot accuracy is reported using one caption embedding prompted by "a photo of". ImageNet is abbreviated INet.

| | Source | Cross-dataset Evaluation Targets | | | | | | | | | | | Domain Generalization Targets | | | | |
|---|---|---|---|---|---|---|---|---|---|---|---|---|---|---|---|---|---|
| | INet | Caltech | Pets | Cars | Flowers | Food | Aircraft | SUN | DTD | EuroSAT | UCF | Average | INet-V2 | INet-Sketch | INet-A | INet-R | Average |
| CLIPood | 76.6 | **97.2** | **94.3** | 92.7 | 77.5 | 87.3 | 37.4 | **74.3** | 60.3 | 59.5 | 75.2 | 75.6 | **69.5** | 64.9 | 56.6 | 88.7 | **69.9** |
| + ICE | **76.7** | **97.2** | 93.9 | **93.1** | **77.8** | **87.4** | **39.6** | **74.3** | **60.8** | **61.8** | **75.7** | **76.2** | 69.2 | 64.9 | **56.7** | **88.8** | **69.9** |
| CoOp | 76.4 | **97.4** | **93.9** | 93.2 | 77.0 | **87.6** | 39.0 | 73.4 | 59.2 | 61.2 | 74.7 | 75.7 | **69.0** | 63.3 | 55.2 | 87.7 | 68.8 |
| + ICE | **76.7** | 97.2 | 93.8 | **93.5** | **77.7** | **87.6** | **40.7** | **73.6** | **60.6** | **62.0** | **75.1** | **76.2** | **69.0** | **63.4** | **55.9** | **88.1** | **69.1** |
| MaPLe | 77.3 | **96.7** | **94.2** | 92.8 | 76.9 | **87.2** | 39.5 | 73.9 | 61.1 | 58.7 | 76.0 | 75.7 | **70.2** | **64.7** | 54.8 | 88.2 | 69.5 |
| + ICE | **77.5** | 96.6 | 94.1 | **93.0** | **77.1** | **87.2** | **41.4** | **74.3** | **61.3** | **59.6** | **76.8** | **76.1** | 70.1 | 64.6 | **55.1** | **88.6** | **69.6** |

Table 2: Comparison with *few-shot baselines* in the *cross-dataset evaluation* setting and the *domain generalization* setting. The model is fine-tuned on ImageNet with three different methods and tested on 15 total target datasets. The average accuracies are calculated separately for the two settings following prior work. In all cases, we observe that evaluating with our ICE method provides consistent improvements. All methods use CoCa ViT-L/14.

alization datasets, each applying a different distribution shift to the source ImageNet dataset. These include an extension of the ImageNet dataset, ImageNetV2 (Recht et al., 2019); a black and white sketches dataset, ImageNet-Sketch (Wang et al., 2019), a naturally adversarial dataset, ImageNet-A (Hendrycks et al., 2019), and a dataset containing different renditions (e.g. cartoons, graffiti, plush objects, etc.) of the ImageNet classes, ImageNet-R (Hendrycks et al., 2020).

## 4.1 ZERO-SHOT CLASSIFICATION

**Baselines.** We consider four existing SOTA methods as zero-shot classification baselines for ICE: (1) a pre-trained CoCa model (Yu et al., 2022) with class embeddings generated using the prompt "a photo of a {}", where "{}"is replaced by the corresponding class (2) a pre-trained CoCa model (Yu et al., 2022) with class embeddings as the centroids of 80 intermediate class embeddings generated using hand-crafted prompts from Radford et al. (2021) (3) a pre-trained CoCa model using large language model (LLM) generated descriptors from (Menon & Vondrick, 2022) (4) a variation of the previous baseline, where we take the centroid of each descriptor embedding for standard zero-shot classification.

**Results.** In Table 1, we observe that stacking ICE achieves consistent improvements of around 0.5% on average across cross-dataset evaluation and domain-generalization evaluation benchmarks, with improvements of up to 3% on datasets like FGVCAircraft and EuroSAT.

## 4.2 FINE-TUNING BASELINES WITH FEW-SHOT LEARNING

Additionally, we consider SOTA methods where a pre-trained model is fine-tuned on 16-shot ImageNet training data. Specifically, the ImageNet training dataset contains 1000 classes with 16 images per class, for a total of 16,000 images in the train dataset.

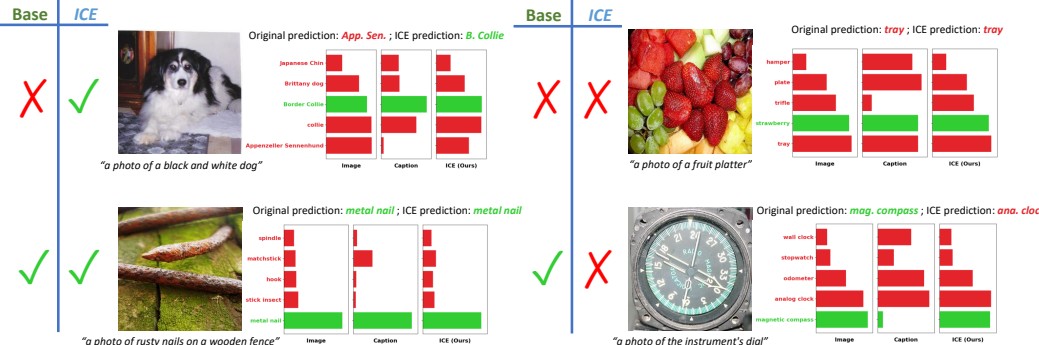

Figure 5: A qualitative analysis on various ways ICE can affect the downstream classification performance.

**Baselines.** We consider two prompt learning methods, CoOp (Zhou et al., 2022b) and MaPLe (Khattak et al., 2023), and a fine-tuning method CLIPood (Shu et al., 2023) as our ICE baselines. We use each method for few-shot fine-tuning on the CLIP components of the CoCa (Yu et al., 2022) architecture, and perform standard zero-shot classification by following the CLIP framework (Radford et al., 2021).

**Results.** As seen in Table 2, applying ICE to each baseline provides improvements of $0.5\%$ on average across cross-dataset generalization evaluation datasets, and smaller improvements for the domain generalization datasets. Importantly, for the methods such as CLIPood where we do not see improvements in domain generalization on average, we find that ICE at least maintains the average performance of the classification backbone.

## 4.3 UNDERSTANDING WHY ICE PROVIDES IMPROVEMENTS

To better understand why ICE improves the base methods paired with it, we analyze examples in Figure 5 where ICE correctly reclassifies a previously incorrectly classified datapoint, where ICE preserves a previously correct classification, where ICE fails to correctly reclassify a previously incorrect classification, and where ICE accidentally incorrectly reclassifies a previously correct classification.

In the top left example of Figure 5, we observe that ICE reclassifies the incorrectly predicted *Appenzeller Sennenhund* class to the correct *Border Collie* class. One intuition for why this occurs is that Border Collies most commonly have black and white fur, whereas Appenzeller Sennenhund dogs typically have unique tricolor fur coats. Thus, the caption "a photo of a black and white dog" would correspond more with the Border Collie purely based on its bicolor fur. Additional examples of correct ICE reclassifications can be found in Figure 1.

In the bottom left example, we observe that ICE is able to successfully preserve an originally correct classification. Here, since the caption agrees with the image predicted class, ICE is able to predict the same class as before.

In the top right image, ICE fails to correctly reclassify an initially incorrect prediction. In this case, the original prediction was a *tray*, which would make sense since this image can technically be described as a "tray of fruits". Similarly, the captions describe the image as "a photo of a fruit platter", which can correspond more to a *plate* or *tray* than a *strawberry*. Thus, the ICE predicted class still matches that of the original class.

Finally, the bottom right image is an example for how ICE can accidentally incorrectly reclassify an initially correct prediction. Here, the correct class is a *magnetic compass*, as evident by the blue text written on the instrument. While the caption "a photo of the instrument's dial" technically describes the image, without additional visual context, it is easy to believe that the caption is describing a clock rather than a magnetic compass since dials tend to be associated with clocks. Due to this ambiguity, ICE was able to convince the initial predicted class to reclassify to the runner-up class of *analog clock*. This example highlights the importance of why we need to consider both the image and caption information for making the most informed decisions.

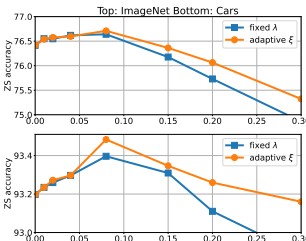 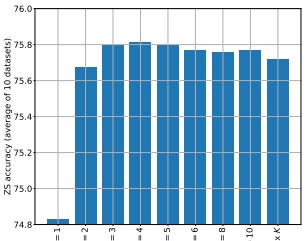 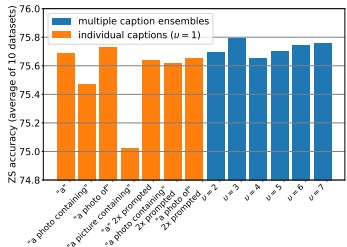

Figure 6: Left: Ablation results on varying $\xi$. Comparison between fixed caption weight $\lambda$ and adaptive $\lambda$ using Eq. 3. Adaptive $\lambda$ is clearly superior on Cars and ImageNet. Center: Ablation results on varying $K$. $K = 4$ is optimal and clearly superior than bypassing Top-$K$ selection step (denoted as max $K$ in bar plot). Right: Ablation results on varying caption prompting and number of captions in ensemble ($v$). $v = 3$ captions is optimal.

## 5 ABLATION STUDIES

A comprehensive ablation study on the parameters of ICE is presented in Figure 6. First, we evaluate the contribution of the adaptive $\lambda$ mechanism proposed in Eq. 3 on the left of the figure. On ImageNet and cars, the zero-shot accuracy of the adaptive $\lambda$ is clearly superior to the fixed $\lambda$ for varying values of $\xi$ (in the adaptive case) and $\lambda$ (in the fixed case). In the middle bar plot, we examine the contribution of the Top-$K$ selection procedure outlined in Eq. 2. We compare multiple values of $K$ (where $K = 1$ is equivalent to ignoring caption embeddings, and "max $K$" denotes score averaging over all classes without Top-$K$ selection). Values of $K$ around $K = 4$ are clearly superior to both ignoring caption embeddings and score averaging without Top-$K$ selection. Finally, in the bar plot on the right, we examine the contribution of ensembling multiple captions. Using $v = 3$ captions is approximately optimal. Note that the 7 individual captions exhibit high variance in zero-shot accuracy (under the ICE framework); this variance in performance is greatly reduced by ensembling a small number of captions.

## 6 LIMITATIONS

While our results show consistent improvements when evaluating on several different baseline methods across a diverse collection of datasets, we note several key limitations of our method. First, as elaborated in Section 3.4, ICE heavily relies on the assumption that the captions can provide useful information about the image that is not fully encoded by the image embeddings. As seen in Section 4.3, when the captions provide unhelpful or adversarial information and there lacks a good selection of caption scores weight $\lambda$, ICE could decrease the base classification performance. Second, determining a good choice for $\lambda$ is non-trivial, as it requires the selector to have an understanding for when to trust the caption or image scores more on a per-datapoint basis. This task is comparable with relevant literature on failure mode prediction using confidence estimation (Tsiligkaridis, 2020; Hendrycks & Gimpel, 2016; Zhu et al., 2023), which is known to be challenging. Finally, generating captions can be expensive, since each additionally generated caption requires an additional forward pass on the base model. For image-text foundation models like CoCa (Yu et al., 2022), which typically contain hundreds of millions of parameters, this strategy can quickly become a time bottleneck for ICE. Thus, it is important to find a balance between the robustness benefits reaped from the number of captions used, and the linearly-increasing time costs for each additional caption generated.

## 7 CONCLUSION

We proposed a novel method for improved zero-shot classification performance based on combining image and caption embeddings. We showed that our method can be easily paired with existing SOTA methods, and provide improvements of $0.5\%$ on average and up to $3\%$ across a diverse array of datasets in cross-dataset generalization and domain generalization. Several ablation studies are presented to study sensitivity of parameters on performance. We performed an in-depth analysis on why our method can help reclassify previously misclassified points, and also cover cases where it might fail. Future work includes extending this work to better balance the weights between image and caption scores, and considering ways to generate more informative captions for improved downstream classification.

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

## APPENDIX A    EXPERIMENT IMPLEMENTATION DETAILS

**ICE.**    For each method ICE is paired with, we simply take the image probability distribution output by our baseline method, and apply ICE with the computed caption scores. The hyperparameters we use to implement ICE in all experiments are $K = 5$, $\xi = 0.08$, $\epsilon = 1 \times 10^{-12}$, $\upsilon = 3$, and $P = \{$"a", "a photo of", "a photo containing"$\}$. We find the best empirical performance when using the centroid of the embeddings of 3 differently prompted captions and dynamically computing the caption scores weight $\lambda$ using Equation 3.

**Baselines.**    We make a good-faith attempt to tune the hyperparameters of each few-shot baseline. We use batch size 64 and SGD with momentum. Training data is sampled in a round robin fashion to maximize class diversity within each mini-batch. CLIPood trains the vision encoder for 750 iterations at learning rate $1 \times 10^{-5}$ with adaptive margin value of 0.1. CoOp trains 3 prompt tokens initialized with "a photo of" for 1250 iterations at learning rate $2 \times 10^{-4}$ with cross-entropy loss. MaPLe trains 3 prompt tokens prepended to each of the first 3 layers on both encoders for 750 iterations at learning rate 1 with cross-entropy loss.

