# OpenReview forum: "ICE: Image-Caption Encoding for Improved Out-Of-Distribution Generalization In Vision-Language Models"
_ICLR.cc/2024/Conference — Submitted to ICLR 2024_

### Official Review · Reviewer_jLvo · 2023-10-25

**Soundness:** 3 good
**Presentation:** 3 good
**Contribution:** 2 fair
**Rating:** 6
**Confidence:** 3

**Summary:**

The paper proposes a method called Image-Caption Encoding (ICE) to improve the out-of-distribution generalization performance of vision-language models on image classification tasks.
ICE is a training-free method that combines information from both image embeddings and text caption embeddings generated by the model to make a more informed prediction at test time. The top-k predictions are weighted from the caption predictions and image predictions for improved classification and correcting any mistakes from the image classifier.
Extensive experiments show ICE provides consistent improvements of around 0.5% on average and up to 3% on challenging datasets when added to existing state-of-the-art baseline methods for both zero-shot classification and few-shot fine-tuning.

**Strengths:**

**Originality**

The idea of using image captions at evaluation time to improve OOD image classification is novel. The approach of combining image and caption probabilities is also creative, building on ideas from ensembling while utilizing unique properties of captions.

**Quality**

The paper presents thorough experiments across 15 datasets with multiple SOTA baselines. Ablation studies analyze the impact of key parameters. The paper also provides examples and analysis to develop an intuition for why and how ICE works, what are it's limitations and when it fails.

**Clarity**

The paper is clearly written and easy to follow. The method is intuitively explained with figures. Experiments and results are well-organized.

**Significance**

Improving out-of-distribution generalization is an important problem. The consistent gains from this simple approach could make the method widely applicable by utilizing informative captions in unique ways to improve OOD classification.

**Weaknesses:**

- As discussed in the limitations, the method relies on captions providing useful supplementary information. Generating captions from a stronger vision-language model (such as BLIP) and then combining those captions with image predictions could help to make caption selection more robust.

- Determining the optimal weight between image and caption probabilities seems challenging. Would a learning-based approach that adaptively learns weights for each branch work better?  Exploring other ways to set this weight adaptively could strengthen the approach and give more insights into how failure can be handled.

- There is no comparison to other ensembling techniques that could provide diversity. While the motivation behind using captions to improve the OOD is interesting, the improvements from the model are small which raises two questions - a) are the captions generated the main problem, or b) are the way they are used to correct the prediction? A more descriptive SOTA captioning model would help in answering the first question and hence lead the way to design better ensembling techniques.

**Questions:**

No specific questions, please look at weaknesses for certain clarifications.

---

> ### Author Response · Authors · 2023-11-18
> **Response to Reviewer jLvo**
>
> **Comment:** As discussed in the limitations, the method relies on captions providing useful supplementary information. Generating captions from a stronger vision-language model (such as BLIP) and then combining those captions with image predictions could help to make caption selection more robust.
>
> **Response:** Thank you for your suggestion, this is an interesting topic of investigation. Here are some supplementary results on ImageNet and 10 other datasets using BLIP-2 captions instead of CoCa captions:
>
> |  |  INet | Caltech | Pets | Cars | Flowers | Food | Aircraft | SUN | DTD | EuroSAT | UCF | Mean |
> | --- | --- | --- | --- | --- | --- | --- | --- | --- | --- | --- | --- | --- |
> | Zero-shot (image) | 75.1 | 97.6 | 93.8 | 92.7 | 77.3 | 87.5 | 36.6 | 73.6 | 57.2 | 58.5 | 73.4 | 74.8 |
> |     + CoCa ICE | 75.6 | 97.0 | 93.5 | 93.0 | 77.6 | 87.6 | 40.0 | 73.9 | 59.8 | 61.1 | 74.3 | 75.8|
> |     + BLIP2 ICE | 75.3 | 97.2 | 93.5 | 92.7 | 77.8 | 87.8 | 36.6 | 74.1 | 59.8 | 60.9 | 75.4 | 75.6 |
>
> We use the largest available BLIP-2 model, which contains a ViT-g image encoder with a FlanT5-XXL LLM. We observe a similar accuracy gain when applying ICE with these BLIP-2 captions. BLIP-2 captions achieve slightly lower numerical results than CoCa captions on average, despite using a much larger model. We think this is because BLIP-2 sacrifices fine-grained information in favor of a more descriptive caption. For example, for one image in the Pets dataset, the CoCa caption is: "a photo of a abyssinian cat'', while the BLIP-2 caption reads: "a photo of a cat standing on a table''. The CoCa caption gives us the correct classification for the pet in the image, while the BLIP-2 caption describes the scene in general terms without providing the information necessary for pets classification. These results illustrate that simply using a better captioner does not necessarily lead to better ZS accuracy under our ICE framework. However, we believe that using LLM prompting techniques to elicit more fine-grained information from large captioning models and incorporating that information into ICE is an interesting direction for future work.
> We will include our analysis with BLIP in our revision as part of the supplementary material.
>
> [blip2] Li, Junnan, et al. "Blip-2: Bootstrapping language-image pre-training with frozen image encoders and large language models." arXiv preprint arXiv:2301.12597 (2023).
>
> **Comment:** Determining the optimal weight between image and caption probabilities seems challenging. Would a learning-based approach that adaptively learns weights for each branch work better? Exploring other ways to set this weight adaptively could strengthen the approach and give more insights into how failure can be handled.
>
> **Response:** Thank you for your suggestion, this is an excellent point.
> We emphasize that ICE is a zero-shot method, i.e. we assume no additional training or access to test-time information.
> However, if we relax the constraints to allow for some learning, there can definitely be room for improvement.
>
> One method we attempted was adaptively learning the value of $\xi$ as a function of image and caption embeddings. $\xi$ determines the weight between the image and caption predictions according to Equation (3). We use a simple Multi-Layer Perceptron with ReLU activations and sigmoid output to predict $\xi$ and train the network on few-shot ImageNet data. We use the same $\xi$ prediction network for all datasets at test time. The results are provided in the tables below:
>
> |  | INet-V2 | Sketch | INet-A | INet-R | DG Mean |
> | --- | --- | --- | --- | --- | --- |
> |  ICE accuracy (fixed $\xi$) | 67.7 | 63.8 | 54.4 | 87.4 | 68.3 |
> | ICE with adaptive $\xi$      | 67.9 | 63.9 | 54.8 | 87.5 | 68.5 |
>
> |  |  INet | Caltech | Pets | Cars | Flowers | Food | Aircraft | SUN | DTD | EuroSAT | UCF | Mean |
> | --- | --- | --- | --- | --- | --- | --- | --- | --- | --- | --- | --- | --- |
> |  ICE accuracy (fixed $\xi$) | 75.6 | 97.0 | 93.5 | 93.0 | 77.6 | 87.6 | 40.0 | 73.9 | 59.8 | 61.1 | 74.3 | 75.8 |
> | ICE with adaptive $\xi$       | 75.7 | 97.4 | 93.7 | 92.9 | 77.7 | 87.2 | 40.2 | 73.7 | 58.4 | 59.3 | 74.6 | 75.5 |
>
> We observe that the adaptive $\xi$ improves accuracy on all the ImageNet datasets, including both the ImageNet validation set and the domain-shifted versions. This improvement is relatively small due to the limited capacity of the $\xi$ network, but it is consistent. This is a promising result that demonstrates that the $\xi$ can be learned. The adaptive $\xi$ degrades the accuracy on some datasets due to the large label-space shift relative to ImageNet classes (which the $\xi$ network was trained on). However, the promising results on the domain generalization targets could warrant further work. We welcome further discussion on alternative methods to find better weights between the image and caption embeddings.

---

> ### Author Response · Authors · 2023-11-18
> **Response to Reviewer jLvo continued**
>
> **Comment:** There is no comparison to other ensembling techniques that could provide diversity. While the motivation behind using captions to improve the OOD is interesting, the improvements from the model are small which raises two questions - a) are the captions generated the main problem, or b) are the way they are used to correct the prediction? A more descriptive SOTA captioning model would help in answering the first question and hence lead the way to design better ensembling techniques.
>
> **Response:** Our method is complementary to other ensembling methods, such as ensembling image augmentations or ensembling multiple pretrained models, because we leverage a different source of diversity.
> From our experiments, we believe that we have performed a comprehensive analysis of our method stacked on top of prior SoTA few-shot OOD methods.
> As an aside, we claim ICE to be different from other ensembling techniques because we do not use other independently trained models as part of our predictions;
> we specifically take advantage of captioning properties inherent in the CoCa architecture to better guide downstream classification.
>
>
> To address whether the captions being generated are the main problem or the way they are being used: the BLIP experiment above confirms that merely more "descriptive'' captions do not increase the zero-shot accuracy.
> However, this was likely because BLIP-2 captions do not contain more information relevant to the classification problem.
> We believe that captions which contain more specific information about objects in the image could increase the performance of zero-shot ICE.
> Hence, automated prompting of large VL models like BLIP-2 to elicit more specific information would be an interesting direction for future work.

---

> ### Comment · Reviewer_jLvo · 2023-11-23
> **Response to reviewers**
>
> Thanks to the authors for the detailed response and new experimental results.
> My main concern is regarding the new BLIP-2 experiments as it does not lead to improvements in performance. As also pointed out by reviewer oC1d, this could be due to the caption itself or LLM prompting. As for fine-grained classes it is hard to get captions that are useful for improving classification accuracy, this method will fall short. Maybe combining this with other sources of knowledge and such would lead to better gains
>
> Given the response from other reviewers and the limitations of the method, I would like to keep the borderline score.

---

### Official Review · Reviewer_DXxh · 2023-10-31

**Soundness:** 3 good
**Presentation:** 3 good
**Contribution:** 2 fair
**Rating:** 3
**Confidence:** 4

**Summary:**

The address the problem of out-of-distribution (OOD) generalization of image classification models. The proposed method: ICE, is build on the premise that when a OOD datapoint is misclassified, the correct class can sometimes be found in the Top-K predicted classes. To take advantage of this property, ICE enforces consistency between the image-conditioned and caption-conditioned predictions at evaluation time. Evaluation shows that Top-1 OOD accuracies improve by 0.5% on average when the proposed ICE framework is used.

**Strengths:**

+ The proposed approach is well-motivated. Section 4.3 provides details of why the proposed ICE method can improve over baseline zero-shot/few-shot classification approaches in certain cases.

+ The paper is well written, Figure 4 provides a good overview of the proposed approach.

+ The paper reports hyper-parameters and training details, which improves reproducibility.

+ The paper includes adequate ablations in Figure 6, discussion the effect of the weight parameters \lambda and \eta.

+ The paper discussion its limitations in detail.

**Weaknesses:**

- Inference time: Compared to prior work such as  CoCa (Yu et al., 2022), the proposed ICE framework needs to encode/decode multiple captions. This would likely significantly increase inference time compared to prior work. A thorough analysis of inference time with respect to prior work is necessary.

-  Performance improvement over baselines is limited. While the average performance improvement is 0.5%, in many datasets the performance improvement over the baseline is less than 0.1%, e.g., Caltech, Food, SUN, UCF in case of zero-shot cross-dataset generalization. Furthermore, in the case of few-shot domain generalization in Table 2 the best performance is obtained by the baseline CLIPood method.

- In-domain performance: the proposed method is evaluated primarily on cross-dataset and domain generalization settings. However, the in-domain performance, e.g., on ImageNet, is not evaluated (\cf Figure 4 in CoCa).

-  The proposed method seems to be applicable only to image datasets. However, prior work such as CoCa is applicable even to video datasets. A discussion on the applicability of the proposed approach to video datasets would be highly appreciated.

- As the approach looks only at the Top-K classes, its performance is inherently limited. Is the proposed approach helpful in case the correct class is not within the initial Top-K predictions?

**Questions:**

1. A detailed analysis of inference speeds with and without the use of the proposed ICE method would be helpful.
2. The paper should discuss in more detail in which scenarios ICE provides a performance boost, as in the case of many datasets, e.g., Caltech, Food, SUN, and UCF, the performance improvement is less than 0.1%.
3.  The paper should also discuss in more detail the applicability of the proposed approach to video data.

---

> ### Author Response · Authors · 2023-11-18
> **Response to Reviewer DXxh**
>
> **Comment:** A detailed analysis of inference speeds with and without the use of the proposed ICE method would be helpful.
>
> **Response:** Thank you, we provide a supplemental analysis on inference speed in this response. This information is provided in the following table. We use a V100 GPU with batch size 32, no gradient tracking, image size 224x224, and the CoCa ViT L/14 model.
>
> | Method | Evaluation throughput (images/second) |
> | --------- | ---------|
> | Zero-shot (image embeddings only) | 65.9 |
> | ICE zero-shot | 16.8 |
>
> ICE evaluation is indeed slower than zero-shot inference with only image embeddings, due to the additional time for generating captions. However, we emphasize that our method does not focus on inference speed, but rather demonstrating a simple method to improve zero-shot accuracy by incorporating captions into the prediction. It is important to weigh the gains in zero-shot accuracy against the additional inference time. Furthermore, there is extensive work in the area of fast LLM inference we can leverage to speed up caption evaluation (such as quantization methods like [smoothquant]). We can also use simple tricks such as decreasing the number of tokens in the generated caption. This is outside the scope of the current work.
>
> [smoothquant] Xiao, Guangxuan, et al. "Smoothquant: Accurate and efficient post-training quantization for large language models." International Conference on Machine Learning. PMLR, 2023.
>
> **Comment:** The paper should discuss in more detail in which scenarios ICE provides a performance boost, as in the case of many datasets, e.g., Caltech, Food, SUN, and UCF, the performance improvement is less than 0.1%.
>
> **Response:** We acknowledge that ICE provides smaller gains on several datasets for few-shot classification in Table 2, especially for domain generalization.
> However, we first note that ICE is a zero-shot method, and thus our gains of 0.5% on average in Table 1 better reflect the capability of our approach.
> We include supplementary few-shot training results in Table 2 to show that even when deploying a zero-shot method to a few-shot context, we are able to enjoy gains of 0.5% on average on cross-dataset generalization.
> Furthermore, on several of the datasets where we fail to see gains, we note that there was little room for improvement to begin with.
> This can be observed in the table we display here.
> One hypothesis that can be made is if the classification accuracy is already very high, ICE may underperform because it is more likely to incorrectly reclassify a previously correct sample than vice versa.
>
> |  |  INet | Caltech | Pets | Cars | Flowers | Food | Aircraft | SUN | DTD | EuroSAT | UCF | Mean |
> | --- | --- | --- | --- | --- | --- | --- | --- | --- | --- | --- | --- | --- |
> | Base accuracy (image zero-shot) | 75.1 | 97.6 | 93.8 | 92.7 | 77.3 | 87.5 | 36.6 | 73.6 | 57.2 | 58.5 | 73.4 | 74.8 |
> | ICE accuracy | 75.6 | 97.0 | 93.5 | 93.0 | 77.6 | 87.6 | 40.0 | 73.9 | 59.8 | 61.1 | 74.3 | 75.8 |
> | Both correct (\%) | 73.4 | 96.8 | 92.5 | 92.1 | 75.4 | 86.3 | 34.1 | 72.1 | 56.6 | 57.7 | 72.0 | 73.6 |
> |  Base incorrect, ICE correct (\%) | 2.2 | 0.2 | 1.0 | 0.9 | 2.2 | 1.3 | 5.9 | 1.8 | 3.1 | 3.4 | 2.3 | 2.2 |
> |  Image correct, ICE incorrect (\%) | 1.7 | 0.8 | 1.2 | 0.6 | 1.9 | 1.2 | 2.5 | 1.5 | 0.5 | 0.8 | 1.4 | 1.2 |
> | Both incorrect (\%) | 22.7 | 2.2 | 5.2 | 6.4 | 20.5 | 11.2 | 57.5 | 24.6 | 39.7 | 38.1 | 24.2 | 23.0 |
>
> We would like to highlight that ICE achieves a 2\% gain over few-shot baselines, and a 3\% gain over zero-shot baselines on the aircraft dataset. This is likely because aircraft classification requires attending to details in the image (such as wingtip styles, numbers on the fuselage etc.). Traditional zero-shot CLIP uses the global image feature for classification, which may ignore these details. On the other hand, the CoCa decoder considers all spatial image tokens. Thus the caption is better at capturing local details. We can clearly see this when asking CoCa to describe an image with writing in it, since the resulting caption usually faithfully transcribes the writing. Therefore, it is intuitive that using ICE to take into account caption information would be beneficial to classification problems where discriminative visual attributes are small. We will include this additional analysis in our revision as part of the supplementary material.
>
> **Comment:** In-domain performance: the proposed method is evaluated primarily on cross-dataset and domain generalization settings. However, the in-domain performance, e.g., on ImageNet, is not evaluated (cf Figure 4 in CoCa).
>
> **Response:** Thank you for pointing this out.
> We have conducted an in-domain evaluation of ImageNet (labeled as *INet*) in Table 1 and Table 2.
> We will reorganize our results before publishing to clarify any confusion.

---

> ### Author Response · Authors · 2023-11-18
> **Response to Reviewer DXxh continued**
>
> **Comment:** The paper should also discuss in more detail the applicability of the proposed approach to video data.
>
> **Response:** Thank you for your suggestion, video data is indeed an important component of the original CoCa paper (Yu et. al., 2022).
> One of the limitations of our method is that it requires additional forward passes on the base model to generate captions, creating a time bottleneck.
> Thus, it would be difficult to scale ICE to video data where we must perform online predictions on a stream of data.
> However, we note that our current focus is to provide gains for zero-shot image classification, and we believe that our method provides a novel perspective on how to correctly pivot toward the correct class.
> Future work can focus on reducing the computational load of our method.
> For example, one potential direction is tuning the captioner to give more descriptive captions (e.g. color of the image, textures, etc.) rather than directly state the objects in the scene.
> If we can generate an informative caption in the first pass, we can reduce the number of captions needed for ICE.
> Another potential direction is detecting change in video data, and only regenerating captions when necessary.
>
> We were not able to perform video classification experiments in the short span of the rebuttal period, but we propose the following more scalable way of extending ICE to video recognition. Captioning each frame of a video is impractical, since tokens must be generated one after another. However, we can consider captioning multiple frames at a time. The CoCa decoder takes as input a spatial token map from the image encoder. We could pool the spatial tokens from multiple video frames before sending them to the text decoder. The authors of CoCa already proposed an attention pooling mechanism to ensemble the spatial tokens from different frames; we could leverage this to generate a video caption.  We can then use ICE to aggregate the predictions based on the pooled video image embedding and the generated caption.
> We welcome additional discussion for how our approach can be extended toward video data.
>
> **Comment:** As the approach looks only at the Top-K classes, its performance is inherently limited. Is the proposed approach helpful in case the correct class is not within the initial Top-K predictions?
>
> **Response:** Correct, the proposed method only allows for pivoting within the Top-$K$ predictions, which means that we are unable to correctly reclassify to far-away classes.
> However, we believe that this is a reasonable assumption to make, since we are motivated by the observation in Figure 2a that in most cases, the correct class can already be found in the Top-$K$ predictions.
> Thus, in those cases, we wish to provide additional information to the model to pivot toward the correct class.

---

> > ### Comment · Reviewer_DXxh · 2023-11-22
> > **Thank you for the rebuttal**
> >
> > I thank the authors for their efforts.
> >
> > The limited applicability of the approach is due to inference time enhancements with captions and the limited improvements in terms of percentage accuracy.
> >
> > I will therefore keep my score.

---

### Official Review · Reviewer_oC1d · 2023-11-01

**Soundness:** 2 fair
**Presentation:** 3 good
**Contribution:** 2 fair
**Rating:** 5
**Confidence:** 4

**Summary:**

The paper proposes to improve the zero-shot and few-shot image classification performance for contrastive vision-language models by utilizing the caption embeddings, given an additional caption model. The proposed method is straightforward and comprehensive evaluations on 11 downstream datasets demonstrates the effectiveness of the method.

**Strengths:**

- The paper is well written and the organization of the paper is clear.
- Experiments are comprehensive with multiple downstream datasets under the zero-shot and few-shot setting.

**Weaknesses:**

- It remains unclear why the text decoder from CoCa is used. It seems that the proposed method only requires a textual description of the input image (any off-the-self high-quality caption models may work). It would be interesting to compare with the performance when using other caption models.

- Compared to CLIP, the approach requires an additional caption model (as the image decoder) and is dependent on the quality of the caption, which can be hard to measure.

- Empirical improvement in the few-shot setting seems marginal compared to the baselines. It may be arguable whether the additional computational cost (of forwarding passing the image to the text decoder) is desirable.

**Questions:**

- Can authors explain why CoCa is used, instead of an arbitrary caption model? If we replace CoCa with SoTA caption model, will the performance be significantly improved?

- The performance gain in the few-shot setting seems marginal (Table 2). Does this mean that in the few-shot adaptation setting, we may just need a few images, instead of using an additional decoder?

---

> ### Author Response · Authors · 2023-11-18
> **Response to Reviewer oC1d**
>
> **Comment:** Can authors explain why CoCa is used, instead of an arbitrary caption model? If we replace CoCa with SoTA caption model, will the performance be significantly improved?
>
> **Response:** Good question. We would first like to reiterate our rationale for choosing CoCa as our base model; then we will present supplementary results with the SoTA BLIP-2 Vision-language (VL) model.
>
> In the context of ICE, CoCa has the following nice properties as compared to other VL models:
>
>  * CoCa is optimized to perform zero-shot classification (same as image-to-text retrieval), while many SoTA VL models such as BLIP-2 and Flamingo place more emphasis on generating human-like responses to queries. Consequently, SoTA VL models do not output a caption that is fine-grained enough for certain classification tasks, at least not without additional prompting.
>
>  * CoCa contains a text encoder, not just a text decoder. This allows us to efficiently perform zero-shot classification with just one model.
>
> We present results on ImageNet and 10 other datasets using BLIP-2 captions instead of CoCa captions in the following table.
>
> |  |  INet | Caltech | Pets | Cars | Flowers | Food | Aircraft | SUN | DTD | EuroSAT | UCF | Mean |
> | --- | --- | --- | --- | --- | --- | --- | --- | --- | --- | --- | --- | --- |
> | Zero-shot (image) | 75.1 | 97.6 | 93.8 | 92.7 | 77.3 | 87.5 | 36.6 | 73.6 | 57.2 | 58.5 | 73.4 | 74.8 |
> |     + CoCa ICE | 75.6 | 97.0 | 93.5 | 93.0 | 77.6 | 87.6 | 40.0 | 73.9 | 59.8 | 61.1 | 74.3 | 75.8|
> |     + BLIP2 ICE | 75.3 | 97.2 | 93.5 | 92.7 | 77.8 | 87.8 | 36.6 | 74.1 | 59.8 | 60.9 | 75.4 | 75.6 |
>
> We use the largest available BLIP-2 model, which contains a ViT-g image encoder with a FlanT5-XXL LLM. We observe a similar accuracy gain when applying ICE with these BLIP-2 captions. BLIP-2 captions achieve slightly lower numerical results than CoCa captions on average, despite using a much larger model. We think this is because BLIP-2 sacrifices fine-grained information in favor of a more descriptive caption. For example, for one image in the Pets dataset, the CoCa caption is: "a photo of a abyssinian cat'', while the BLIP-2 caption reads: "a photo of a cat standing on a table''. The CoCa caption provides the correct classification for the pet in the image, while the BLIP-2 caption describes the scene in general terms without providing the information necessary for pets classification. These results illustrate that simply using a better captioner does not necessarily lead to better ZS accuracy under our ICE framework. However, we believe that using LLM prompting techniques to elicit more fine-grained information from large captioning models and incorporating that information into ICE is an interesting direction for future work. It would be very helpful for us to hear your thoughts on our analysis here.
>
> [blip2] Li, Junnan, et al. "Blip-2: Bootstrapping language-image pre-training with frozen image encoders and large language models." arXiv preprint arXiv:2301.12597 (2023).
>
>
>
> **Comment:** The performance gain in the few-shot setting seems marginal (Table 2). Does this mean that in the few-shot adaptation setting, we may just need a few images, instead of using an additional decoder?
>
> **Response:** Thank you for pointing this out. From Table 2, we observe that the Domain Generalization (DG) targets have the lowest average gain; the Cross-Dataset (CD) Evaluation targets still enjoy 0.5% performance boosts on average.
> However, we still see consistent improvements on hard DG datasets such as ImageNet Adversarial (INet-A) and ImageNet Rendition (INet-R).
> We note that similar trends can be found in the zero-shot data in Table 1, and thus believe that it is more important to evaluate the importance of using an additional decoder based on the properties of the target dataset (as elaborated in Section 3.4 Caption Properties).
> In addition, we note that ICE is designed to be a zero-shot method, making Table 1 results more relevant to our target domain, but we think it is promising that ICE can provide benefits even in a few-shot context.

---

> ### Comment · Reviewer_oC1d · 2023-11-22
> **Thanks for the detailed response**
>
> Thank you for providing detailed responses and the additional experiments with BLIP-2.
>
> It is interesting to see that a better captioning model such as BLIP-2 does not necessarily lead to superior zero-shot classification performance. It remains a bit unclear to me if is it because the fine-grained information from large captioning models is not accurate or the language encoder cannot properly handle such information. This brings some further questions such as what properties are desirable from image captions for zero-shot visual classification and how to better exploit a general caption model beyond utilizing a specific pre-trained model such as CoCa? For example, does BLIP-2 really do not understand what an abyssinian cat is? I appreciate the authors' efforts in showing this addition results and it is a good starting point for answering one of the core scientific questions of this work.
>
>
> I understand that the motivation for using CoCa is that it allows authors to "efficiently perform zero-shot classification with just one model". However, this framework can be interpreted as both an advantage (simplicity) and disadvantage (e.g., post-hoc adjustment, strong reliance on CoCA and therefore would be difficult to generalize to different domains or general visual reasoning tasks).
>
> As for efficiency, the proposed method uses three components: image encoder -> text decoder -> text encoder (as shown in Figure 3). Alternatively, if we use simple vision-language models such as LLaVA [1], we still have the same number of components: image encoder (in LLAVA) -> text decoder (in LLAVA) -> text encoder (in CLIP). Therefore, it might be hard to argue which one is more efficient.
>
> Given the above considerations, other reviewers' comments, and the empirical performance, my judgement for the work remains borderline (either marginally below or above the acceptance threshold).
>
> [1] Liu et al., Visual Instruction Tuning, NeurIPS 2023

---

### Official Review · Reviewer_F7uJ · 2023-11-06

**Soundness:** 2 fair
**Presentation:** 3 good
**Contribution:** 2 fair
**Rating:** 5
**Confidence:** 4

**Summary:**

The authors observed that even when image classification models make a mistake, the correct answer frequently appears within the top five guesses. Inspired to nudge the model towards the right choice among these top contenders, they introduced a novel approach for zero-shot image classification by combining image and text caption embeddings. This technique enhances the performance of the SOTA methods by an average of 0.5%, and by as much as 3%. They have tested their method on cross-dataset generalization datasets and domain generalization datasets. They also performed simple qualitative error analysis.

**Strengths:**

- Tested their method using a wide range of image recognition datasets (11 cross-dataset generalization datasets and 4 domain generalization datasets)
- Ablation studies on the parameters of ICE is performed.

**Weaknesses:**

- The performance gains are slight; examining Table 1 and Table 2 shows that ICE only slightly increases performance over the baseline by 0.1 to 0.4 on variations of ImageNet. Moreover, for certain datasets like INet-Sketch, the baseline actually performs better.
- Image captioning poses a greater challenge compared to image classification because, particularly for complex images (including some found in ImageNet) a single image may not correspond to just one correct label. Consider, for instance, the top-right picture in Figure 5 labeled as "strawberry." Labeling this image solely as a strawberry seems inaccurate. More broadly, it's impractical to use images featuring multiple objects for a single-label classification. Image captioning is the suitable approach for these types of images. Pushing for precise classification on such images may simply be tailoring models to fit benchmark datasets without truly enhancing their comprehension of the images. Therefore, I don't see how their approach can be applied for complex images. I agree that this approach works for simple images, but I'd assume the existing image models already work well for such cases, as evidenced by Table 1 and 2.
- In Section 4.3, "Understanding Why ICE Provides Improvements," the explanation is merely qualitative, based on just four examples. It's difficult to gauge the representativeness of these cases. A deeper quantitative analysis—for example, determining how common these cases are in standard image datasets—would be necessary to fully understand why and how ICE is effective.

**Questions:**

Related to the last point in the Weaknesses section, how representative are four scenarios discussed in Section 4.3 within standard image datasets?

---

> ### Author Response · Authors · 2023-11-18
> **Response to Reviewer F7uJ**
>
> **Comment:** Image captioning poses a greater challenge ...
>
> **Response:** Thank you for providing valuable insight into this problem.
> We agree that image captioning poses a greater challenge than image classification, and captions can describe several objects in the scene that may conflict with the single-label classification objective.
> We first note that if multiple objects exist in the image which correspond to different labels in the dataset, then it is likely a poor problem formulation for single-label image classification to begin with.
> Thus, dealing with these conflicting scenarios would be out of the scope of our method.
> Second, our method works only to pivot toward the correct class in the top-$K$ predicted classes.
> This means that even if the caption provides descriptions about unrelated objects, we can only select a class that was already predicted with high probability by the original image encoder.
> The combination of our image and caption embeddings ensures that we will not stray too far from the initial prediction for image classification which provides a useful constraint.
> Finally, Tables 1 and 2 show that our method can give good zero-shot improvements to existing state-of-the-art models.
>
> **Comment:** In Section 4.3, "Understanding Why ICE Provides Improvements," the explanation is merely qualitative, based on just four examples. It's difficult to gauge the representativeness of these cases. A deeper quantitative analysis—for example, determining how common these cases are in standard image datasets—would be necessary to fully understand why and how ICE is effective.
>
> **Response:** Thank you for suggesting a quantitative analysis of the four situations presented in Section 4.3 and Fig. 5; we agree that this enhances our discussion. Accordingly, we provide the quantity of images in each category as a percentage of total test images in the tables below, for each dataset. In all the datasets, most images are either correctly predicted by both methods or incorrectly predicted by both methods. This is expected, since every dataset contains a large amount of easy images and a large amount of impossible images given a constant model capacity. However, we do observe that in all datasets except Caltech and Pets, the percentage of images where ICE successfully reclassifies an initially incorrect prediction exceeds the percentage of images where ICE incorrectly reclassifies an initially correct prediction. Furthermore, the percentage of failed re-classifications is small when compared to the percentage of successful re-classifications on datasets such as DTD (0.5\% compared to 3.1\%) and EuroSAT (0.8\% compared to 3.4\%). Finally, we note that images incorrectly classified by ICE may be images that are hard to classify without additional context or ambiguous images.
> Future work can help identify which samples to apply ICE to decrease the number of correct-to-incorrect ICE re-classifications, and we welcome further discussion about alternative directions.
> We will include this additional analysis in our revision as part of the supplementary material.
>
> |  |  INet | Caltech | Pets | Cars | Flowers | Food | Aircraft | SUN | DTD | EuroSAT | UCF | Mean |
> | --- | --- | --- | --- | --- | --- | --- | --- | --- | --- | --- | --- | --- |
> | Base accuracy (image zero-shot) | 75.1 | 97.6 | 93.8 | 92.7 | 77.3 | 87.5 | 36.6 | 73.6 | 57.2 | 58.5 | 73.4 | 74.8 |
> | ICE accuracy | 75.6 | 97.0 | 93.5 | 93.0 | 77.6 | 87.6 | 40.0 | 73.9 | 59.8 | 61.1 | 74.3 | 75.8 |
> | Both correct (\%) | 73.4 | 96.8 | 92.5 | 92.1 | 75.4 | 86.3 | 34.1 | 72.1 | 56.6 | 57.7 | 72.0 | 73.6 |
> |  Base incorrect, ICE correct (\%) | 2.2 | 0.2 | 1.0 | 0.9 | 2.2 | 1.3 | 5.9 | 1.8 | 3.1 | 3.4 | 2.3 | 2.2 |
> |  Image correct, ICE incorrect (\%) | 1.7 | 0.8 | 1.2 | 0.6 | 1.9 | 1.2 | 2.5 | 1.5 | 0.5 | 0.8 | 1.4 | 1.2 |
> | Both incorrect (\%) | 22.7 | 2.2 | 5.2 | 6.4 | 20.5 | 11.2 | 57.5 | 24.6 | 39.7 | 38.1 | 24.2 | 23.0 |
>
>
> |  | INet-V2 | Sketch | INet-A | INet-R | DG Mean |
> | --- | --- | --- | --- | --- | --- |
> | Base accuracy (image zero-shot) | 67.5 | 63.5 | 53.8 | 87.0 | 67.9 |
> | ICE accuracy | 67.7 | 63.8 | 54.4 | 87.4 | 68.3 |
> | Both correct (\%) | 65.5 | 61.6 | 51.8 | 86.3 | 66.3 |
> |  Base incorrect, ICE correct (\%) | 2.2 | 2.2 | 2.6 | 1.1 | 2.0 |
> |  Image correct, ICE incorrect (\%) | 1.9 | 1.9 | 2.0 | 0.7 | 1.7 |
> | Both incorrect (\%) | 30.4 | 34.3 | 43.6 | 11.9 | 30.0 |

---

> > ### Comment · Reviewer_F7uJ · 2023-11-23
> > **Response**
> >
> > Thank you for providing the quantitative analysis for the third point. And I appreciate your explanation regarding the second point. While I understand that this method deals with single-object image settings, since the benefits of using this method in practice seems limited (as I clarified in my first point in the weaknesses section), I maintain my score.

---

### Meta-Review · Area_Chair_zgCo · 2023-12-07

**Metareview:**

**Summary**

With an observation that image classification's top-5 accuracy is often significantly higher than the top-1 acc in the out of distribution (OOD) settings, the authors propose ICE to improve OOD generalization. More specifically, they use CoCa to generate captions from the input image, and then separately encode the image and captions into embeddings. Next, the top-K predictions from the two modality are combined to make the final prediction in a zero-shot fashion. The proposed method demonstrate slight improvements on a wide range of image classification datasets.


**Strengths**
- tested on many image recognition datasets
- Multiple reviewers think this paper is well-written and easy to understand.
- Detailed ablation study is conducted.
- Some often-omitted details such as hyper-parameters, training details, and limitations are properly included.
- One reviewer thinks using captioning to improve OOD classification is novel, and OOD generalization is an important research problem. The reviewer also appreciates the idea of combining probabilities from image and text, which achieves an ensemble effect.

**Weaknesses**
- All reviewers agree that accuracy again is small.
- The additional captioning overhead could be significant. A reviewer point out that this overhead is disproportional to the accuracy gain, and all reviewers agree that the trade-off seems not worth it.
- The proposed method is limited to simple image recognition task (classification on single-object image). It does not work for complex images containing multiple visual objects.
- One reviewer mention that the accuracy upper bound of the proposed method is limited to the base image model's top-K accuracy. In my opinion, this may indicate that working on the base model is the more fundamental direction. The small accuracy gain of ICE makes me think that the proposed direction would not have much impact. The accuracy gain could even diminish when the base model becomes more capable.
- The method is built on CoCa. However, ICE can't work on videos despite CoCa can. This is a huge limitation.
- Lack of comparison with other ensembling techniques.
- Why ICE improves is based on qualitative samples. Quantitative experiments are added in rebuttal, hence partially addressed this issue.
- Optimal weight between image and caption predictions could be difficult to obtain. This is partially addressed during rebuttal by learning it in a low-shot setting.

**Justification For Why Not Higher Score:**

Three out of four reviewers lean negative for this submission. The common issue is that captioning brings a large computation overhead and the respective accuracy gain of ICE is too small. All reviewers agree with this in the AC-reviewers private discussion. The reviewer giving positive rating (6) also acknowledge this weakness. As an AC, I agree that for a relatively simple task (image classification), generating caption is indeed an unnecessary large overhead at inference time. Therefore, the application of ICE would be very limited. My suggestion to improve this work is to find other visual application that requires similar computation time to captioning. If ICE can also improve in this scenario, the inference computation overtime can be better justified.

**Justification For Why Not Lower Score:**

N/A

---

### Decision · Program_Chairs · 2024-01-16

Reject